# OpenReview forum: "Cross-Embodiment Robot Foundation World Models with Latent Actions"
_ICML.cc/2026/Conference — ICML 2026 regular_

### Official Review · Reviewer_23Zn · 2026-03-05

**Soundness:** 3
**Presentation:** 3
**Significance:** 3
**Originality:** 3
**Overall Recommendation:** 4
**Confidence:** 3

**Summary:**

The paper addresses the challenge of training robot world models that can generalize across diverse robot embodiments with heterogeneous action spaces. The authors propose the Latent Action Conditioned Robot World Model (LAC-WM). Instead of conditioning on explicit, embodiment-specific motion labels (which creates disjoint action spaces), LAC-WM learns a unified latent action space using an Inverse Dynamics Model (IDM) and a Motion Decoder during pretraining on multi-embodiment data (including human videos). For adaptation to unseen robots, an action projector is trained to map raw actions into this pre-trained latent space. Experiments demonstrate that LAC-WM significantly outperforms a baseline using explicit action encoding (EAC-WM) in both video prediction quality and downstream manipulation tasks (up to 46.7% improvement), showing positive scaling with the number of pretraining embodiments.
#####

**Compliance With Llm Reviewing Policy:**

Affirmed.

**Final Justification:**

I am satisfied that my main concerns have been addressed.

**Key Questions For Authors:**

- Topological Diversity: How does the unified latent space handle embodiments with fundamentally different topologies? Does the latent space collapse or segregate if the kinematics are too distinct?
- Role of Human Data: In the ablation or analysis, how critical is the inclusion of human data (EgoDex) for the alignment of the two robot datasets (Agibot and Droid)? Would the latent spaces of Agibot and Droid align without the "bridge" of human manipulation data?
- Inference Latency: Does the two-stage process at inference time (Action Projector -> FDM) introduce any significant latency compared to direct explicit conditioning, especially for real-time control loops?
- Long-horizon Prediction: The evaluation looks at 8-frame prediction. How does the model perform over longer horizons? Does the latent action conditioning suffer from drift faster or slower than explicit actions?

**Limitations:**

yes

**Strengths And Weaknesses:**

- Strengths:
  - Significance: The problem of cross-embodiment generalization is a central challenge in robot learning. The paper proposes a scalable solution by abstracting actions into a unified latent space, which is highly relevant to the development of "foundation models" in robotics.
  - Soundness (Methodology): The architecture is well-designed. The use of an Inverse Dynamics Model (IDM) coupled with a Motion Decoder to enforce semantic meaning in the latent space is a sound approach. The strategy of using an "action projector" for fine-tuning allows the heavy backbone (FDM) to remain largely stable while adapting to new action spaces.
  - Empirical Results: The results are compelling. The comparison between LAC-WM and EAC-WM (Explicit Action Conditioned) clearly highlights the limitations of disjoint action spaces. The UMAP visualizations (Figure 2) and cross-embodiment action transfer examples (Figure 3) provide strong qualitative evidence supporting the quantitative gains.
  - Presentation: The paper is clearly written. Figure 1 effectively summarizes the pipeline, and the distinction between pretraining and fine-tuning phases is easy to follow.
- Weaknesses:
  - Baselines: The primary baseline is EAC-WM (a variant constructed by the authors). While logical, it would be beneficial to compare against other state-of-the-art methods that handle cross-embodiment data, such as Open X-Embodiment policies (though they are policies, not world models, a comparison on the downstream task could be relevant context) or other latent action approaches.
  - Action Space Complexity: The evaluation focuses on a bimanual Franka setup (BFA). While complex, it is still a rigid robot arm. It is unclear how well the "unified latent space" holds up if the embodiments are drastically different topologically (e.g., a quadruped robot vs. a manipulator vs. a drone).
  - Dependency on Human Data: The method relies heavily on human hand data (EgoDex) to bridge robot embodiments. The paper could better clarify if the method still works effectively if human data is removed or significantly reduced.

---

> ### Author Rebuttal · Authors · 2026-03-31
>
> We thank the reviewer for the thoughtful feedback.
>
> **W1: Baseline choice.**
> Because the paper’s central question is whether a unified latent action space improves world-model learning from cross-embodiment data, we believe EAC-WM is the most appropriate baseline. It keeps the same overall world-model setting while changing the action-conditioning formulation from latent-action to explicit-action conditioning (Sec. 4.1). This makes the comparison targeted and controlled.
>
> **W2 & Q1: Topological diversity.**
> We thank the reviewer for this insightful question. We agree that evaluating more drastically different embodiment topologies is an important direction for future work. While we currently do not have the data or compute resources to study fundamentally different morphologies such as quadrupeds or drones, the embodiments used in our paper already provide some meaningful variation. In particular, Droid is a single-arm manipulator, whereas Agibot is a bimanual mobile humanoid, and this comparison offers an initial indication of how the latent space behaves across embodiments with different kinematic structures.
>
> As shown in Fig. 2, the latent actions of Agibot and Droid occupy a shared latent space, but with some offset between cluster centers. We believe this is expected and desirable: the shared space captures common action semantics across embodiments, while the offsets preserve embodiment-specific information needed for the world model to generate motions consistent with each platform’s kinematics. Therefore, rather than collapsing or fully segregating, the latent space appears to support a unified structure that still retains embodiment-dependent constraints. Based on this observation, we expect that more topologically different embodiments would also lie in a common latent space, likely with larger cluster offsets reflecting their greater morphological differences.
>
> **W3 & Q2: Role of human data.**
> We believe that one important advantage of the proposed approach is precisely that it can leverage human data, which is a scalable, diverse, and relatively inexpensive source of manipulation experience. More broadly, however, the core advantage of LAC-WM is its ability to learn effectively from cross-embodiment data. As shown in Fig. 5, LAC-WM benefits from adding another robot embodiment during pretraining, whereas EAC-WM degrades when an additional robot embodiment is introduced. This suggests that the benefit of LAC-WM is not limited to the inclusion of human data, but comes from its ability to learn a unified latent action space across heterogeneous embodiments more generally.
>
> We agree that a dedicated ablation on the role of EgoDex would be valuable. Due to compute limitations, we have not yet performed an experiment that removes or significantly reduces the human data, but we hope to investigate this more systematically in future work.
>
> **Q3: Inference latency.**
> The action projector is a lightweight MLP with comparable size to the action encoder used in EAC-WM, so it does not introduce meaningful additional inference overhead. As reported in Appendix A.4, on an NVIDIA L40S GPU, the inference time is approximately 0.1 seconds for a batch size of 10 when predicting 8 future frames at 256 × 256 resolution. In practice, this suggests that using the action projector together with the FDM does not create a significant latency disadvantage compared to explicit-action conditioning.
>
> **Q4: Long-horizon prediction.**
> Both LAC-WM and EAC-WM are trained to predict 8 future steps. In principle, both models can be rolled out autoregressively by feeding previously predicted frames back into the model, but in practice we observe prediction drift for both methods under longer rollouts. For this reason, we focus on 8-step prediction in the current paper. We agree that a more detailed study of long-horizon prediction behavior would be valuable, and we leave this to future work.

---

> > ### Author Rebuttal · Reviewer_23Zn · 2026-04-02
> >
> > Thank you for the thoughtful rebuttal. I am satisfied that my main concerns have been addressed.

---

> > > ### Author Response · Authors · 2026-04-03
> > >
> > > We thank the reviewer for the positive feedback! We have added an additional evaluation on 5 Libero tasks. Because the original Libero benchmark is close to saturated, we construct 5 more challenging variants by introducing unseen objects (book, white bowl, wine bottle) into the basket, or by changing the locations of seen objects (alphabet soup, salad dressing). For action selection, we use a pi0.5 policy trained with 50-step prediction as the VLA. At each decision step, we sample 20 candidate action sequences from the VLA and execute the first 10 steps. For each task, we evaluate 25 rollouts. The results are shown below.
> > >
> > > | Target Object | VLA Random| VLA Average| EAM | LAM |
> > >   |------|-----------------|-----------------|-----|-----|
> > >   | book | 76% | 76% | 80% | **92%** |
> > >   | white_bowl | 76% | 92% | 92% | **96%** |
> > >   | wine_bottle | 72% | 84% | 84% | **96%** |
> > >   | alphabet_soup | 80% | 72% | 80% | **88%** |
> > >   | salad_dressing | 68% | 84% | 76% | **88%** |
> > >   | **Average** | 74.4% | 81.6% | 82.4%| **92.0%** |
> > >
> > > As shown above, LAM consistently outperforms both EAM and the VLA baselines, including VLA with averaged actions. EAM provides only a modest improvement over the VLA baselines, whereas LAM yields a substantial gain, achieving an average success rate of 92.0%.
> > >
> > > We hope these new experiments can further enhance the paper and we would really appreciate it if you can consider increasing your score.

---

### Official Review · Reviewer_ABGF · 2026-03-11

**Soundness:** 3
**Presentation:** 3
**Significance:** 2
**Originality:** 2
**Overall Recommendation:** 3
**Confidence:** 4

**Summary:**

This paper introduces LAC-WM, a robot world model that learns a unified latent action space from diverse embodiment data (human, humanoid, single-arm). This shared space enables better generalization to unseen robots compared to conditioning on raw actions (EAC-WM baseline). Experiments show LAC-WM achieves 46.7% higher task success in downstream planning and scales positively with more embodiments, while EAC-WM degrades.

**Compliance With Llm Reviewing Policy:**

Affirmed.

**Key Questions For Authors:**

1. Your UMAP (Fig. 2) shows clusters remain somewhat separated by embodiment even with motion decoder (Droid vs. Egodex cluster offsets). Is this separation desirable (encoding embodiment-specific execution constraints) or undesirable (incomplete unification)? How would you disentangle whether the motion decoder enforces shared action semantics versus embodiment-specific motion patterns?

2. You select top-k action sequences by lowest δf (embedding distance). Have you compared this against alternatives—e.g., ensemble disagreement, action likelihood under the VLA prior, or random selection?

**Limitations:**

yes

**Strengths And Weaknesses:**

### Strength

1. The controlled experiments in Fig. 5 with fixed data/increasing embodiments and fixed embodiments/increasing data provide strong evidence for the central hypothesis: LAC-WM scales positively, EAC-WM degrades with more embodiments.

2. Using downstream planning success (S.R., S.R.L.) as primary metrics, which demonstrates the practicability of world models in the field of robotics.

3. This paper is easy to read and the implementation details are well stated.

### Weaknesses


1. Despite 22 object instances and categories, the core task is pick-and-place. This is relatively simple compared to the breadth of manipulation tasks. Demonstrating on more diverse tasks (e.g., tool use, assembly) would strengthen significance claims. Also, the 46.7% relative gain sounds impressive, but moving from ~0.15 to ~0.22 success rate still leaves most tasks failing.

2. Without real-robot experiments, it's unclear how the approach transfers from simulation to physical systems, especially given distribution shifts in vision and dynamics.

3. δf depends entirely on the choice of image encoder (here, V-JEPA-2). The paper does not ablate across different encoders (DINOv2, CLIP, etc.) to show δf correlates consistently with task success regardless of encoder choice.

4. IDMs, FDMs, latent action spaces, and motion decoders have all appeared separately in prior work (UniSkill, VILLA-X). Action conditioned world model using latent action representations is also explored in AdaWorld.

---

> ### Author Rebuttal · Authors · 2026-03-31
>
> We thank the reviewer for the thoughtful feedback.
>
> **W1: Limited task diversity and low absolute success.**
> While the current benchmark focuses on pick-and-place, the evaluation is still nontrivial. It involves dexterous manipulation, generalization to both unseen object instances and unseen categories, and action selection on top of a base VLA policy trained from a relatively limited dataset. These factors make the task challenging and help explain the low absolute success rates. We are also actively working on evaluating the approach on other tasks.
>
> **W2: Lack of real-robot experiments.**
> The goal of this paper is not to address sim-to-real transfer, but to study whether a unified latent action space is beneficial for cross-embodiment robot world modeling. For this reason, we focus on controlled evaluation in RoboCasa, which provides diverse and realistic simulated manipulation environments. In addition, as shown in the supplementary action transfer videos, LAC-WM is able to generate realistic imagined rollouts on real datasets such as Droid, EgoDex, and Agibot.
>
> **W3: Dependence of embedding-based metrics on the image encoder.**
> We acknowledge that the absolute value of task performance metrics based on image embeddings may vary with the choice of encoder. However, improving the action selection metric itself is not the focus of this paper. Our main claim is that LAC-WM is a stronger cross-embodiment world model than EAC-WM. We believe this is sufficiently supported by the combination of improved video generation quality in Table 1 and improved downstream task success in Table 2 under the same evaluation setup. Since both methods use the same image encoder, the comparison remains controlled and fair.
>
> **W4: Novelty relative to prior work.**
> We agree that the individual modules used in LAC-WM are related to prior work. However, LAC-WM builds a practical and scalable framework for robot world modeling using these components in a cross-embodiment setting, together with a comprehensive empirical analysis. The contribution of the paper is therefore not a new standalone module, but the overall framework and system design. Experiments on both video generation quality and downstream planning show that LAC-WM consistently outperforms EAC-WM and, importantly, scales better as embodiment diversity increases. We believe this demonstrates a promising and practical solution for learning robot world models from heterogeneous embodiment data.
>
> **Q1: Residual embodiment separation in the latent space.**
> We believe this separation is expected and desirable. Embodiment-specific constraints are necessary for the world model to generate videos that respect the embodiment’s kinematics and execution patterns. In our framework, the latent action encoder encourages shared action semantics across embodiments, while the motion decoder helps preserve the physically meaningful, embodiment-specific motion patterns needed for accurate generation. This leads to a unified latent action space without removing all embodiment-specific structure, as suggested by the UMAP visualization in Fig. 2 and the action transfer results in Fig. 3.
>
> In our current setting, we do not explicitly disentangle shared action semantics from embodiment-specific motion patterns, since the motion decoding loss is optimized jointly with the other objectives. One possible future direction would be to pretrain LAC-WM without motion decoding and then add the motion decoding loss during finetuning, or to introduce a morphology token concatenated with the latent action representation.
>
> **Q2: Alternatives to embedding-distance-based action selection.**
> We use embedding distance because it provides a simple way for the world model to select the action sequence that is most likely to complete the task. We did compare against simpler alternatives in Table 2. Specifically, VLA-random corresponds to random action selection, and VLA-mean averages the sampled actions, which may partially reflect the underlying action prior.

---

> > ### Author Rebuttal · Reviewer_ABGF · 2026-04-06
> >
> > I thank the authors for their thorough response. The controlled comparisons in Tables 1–2 are appreciated, and the framework direction is interesting. However, my main concerns remain partially unresolved.
> >
> > W1–W2: While the generalization to unseen instances/categories is nontrivial, the evaluation is limited to a single manipulation primitive, and ~22% success suggests room for improvement. Real-world closed-loop results would considerably strengthen the "practical framework" claim beyond the simulated video rollouts shown.
> >
> > W4: I accept the systems-level framing, though such contributions benefit from broader empirical validation.
> >
> > I maintain my current score

---

> > > ### Author Response · Authors · 2026-04-07
> > >
> > > We thank the reviewer for the feedback.
> > >
> > > To further address the concern, we added an additional evaluation on 5 Libero tasks. Because the original Libero benchmark is close to saturated, we construct 5 more challenging variants by introducing unseen objects (book, white bowl, wine bottle) into the basket, or by changing the locations of seen objects (alphabet soup, salad dressing). For action selection, we use a pi0.5 policy trained with 50-step prediction as the VLA. At each decision step, we sample 20 candidate action sequences from the VLA and execute the first 10 steps. For each task, we evaluate 25 rollouts. The results are shown below.
> > >
> > > | Target Object | VLA Random| VLA Average| EAM | LAM |
> > >   |------|-----------------|-----------------|-----|-----|
> > >   | book | 76% | 76% | 80% | **92%** |
> > >   | white_bowl | 76% | 92% | 92% | **96%** |
> > >   | wine_bottle | 72% | 84% | 84% | **96%** |
> > >   | alphabet_soup | 80% | 72% | 80% | **88%** |
> > >   | salad_dressing | 68% | 84% | 76% | **88%** |
> > >   | **Average** | 74.4% | 81.6% | 82.4%| **92.0%** |
> > >
> > > As shown above, LAM consistently outperforms both EAM and the VLA baselines, including VLA with averaged actions. EAM provides only a modest improvement over the VLA baselines, whereas LAM yields a substantial gain, achieving an average success rate of 92.0%. Since these Libero tasks are easier than the RoboCasa setting, the strong performance here suggests that the relatively lower absolute success rate of LAM in RoboCasa is primarily constrained by the underlying VLA performance rather than by the quality of the learned world model itself.
> > >
> > > We hope these new experiments help address the concern about the limited evaluation scope. We would also like to emphasize that the main contribution of this paper is the world model training framework. Planning performance is only one of the three evaluation axes we use to assess world model quality, alongside video generation quality and latent space analysis. We sincerely hope the reviewer will re-evaluate the paper in light of this broader perspective.

---

### Official Review · Reviewer_8Mxt · 2026-03-14

**Soundness:** 2
**Presentation:** 3
**Significance:** 3
**Originality:** 2
**Overall Recommendation:** 4
**Confidence:** 4

**Summary:**

LAC-WM is a cross-embodiment robot foundation world model that addresses the limitations of explicit action conditioning, which often leads to disjoint action spaces and poor scaling across different robots.

Instead of relying on raw action labels, LAC-WM utilizes an Inverse Dynamics Model (IDM) to extract visual motion-based latent representations from video sequences.

The architecture is trained in three stages, using a motion decoder and reconstruction objectives to ensure the latent space is physically grounded and unified across diverse embodiments.

The authors evaluate LAC-WM on the RoboCasa/BFA setup, demonstrating improvements over explicit-action baselines in both video prediction quality and VLA-guided planning success rate.

**Compliance With Llm Reviewing Policy:**

Affirmed.

**Key Questions For Authors:**

Please refer to the Weaknesses section above.

**Limitations:**

The authors provide a reasonable discussion of the limitations of this work.

**Strengths And Weaknesses:**

Strengths:

1. The manuscript is well-structured and easy to follow.
The authors present a clear problem formulation, a coherent method description, and a thorough experimental evaluation.
The architecture diagram (Figure 1) effectively illustrates the pretraining and finetuning pipeline, making the overall framework readily accessible to the reader.

2. The paper offers a compelling solution to the cross-embodiment action space heterogeneity problem by constructing a unified latent action space.
By operating in a learned latent space rather than relying on explicit motion labels, the model circumvents the performance degradation typically observed when scaling across robots with incompatible control dimensionalities.

3. The authors provide a comprehensive analysis of the learned latent space through both qualitative and quantitative lenses.
UMAP/PCA visualizations and cross-embodiment video rollouts confirm the semantic coherence of the latent space.
And scaling experiments show consistent performance gains as more pretraining embodiments are incorporated.

Weaknesses:

1. The reported improvements, while consistent, appear incremental in certain scenarios. Moreover, the primary comparisons are limited to VLA-only baselines, which may not be fully convincing. The absence of comparisons with other specialized cross-embodiment architectures leaves the relative merits of the proposed approach insufficiently established.

2. Despite the stated goal of cross-embodiment generalization, adapting the model to a new embodiment still requires a cumbersome three-stage finetuning procedure, which undermines the claimed efficiency of the framework.

3. The need to modify the unified latent space during adaptation to a new robot raises concerns. This process risks compromising the integrity of the original latent representations, potentially eroding the general knowledge acquired during large-scale pretraining.

4. In the experimental setup, EAC-WM employs separate action encoders for each embodiment, meaning its action embedding space is disjoint by construction. This renders the finding that EAC-WM produces a fragmented latent space somewhat circular and weakens the persuasiveness of the comparison.

5. In the ablation study, Table 3 removes both the motion decoder and cross-augmentation simultaneously. This conflates two independent design choices, making it impossible to disentangle their individual contributions to the overall performance.

---

> ### Author Rebuttal · Authors · 2026-03-31
>
> We thank the reviewer for the constructive feedback.
>
> **W1: Incremental gains and baseline choice.**
> We agree that the absolute gains are moderate in some settings, but we believe this is largely because the base VLA policy itself has limited success on this challenging dexterous manipulation benchmark. Importantly, our main question is whether a unified latent action space improves cross-embodiment world modeling, so we compare against EAC-WM, which isolates the choice of latent-action versus explicit-action conditioning while keeping the overall world-model setting aligned.
>
> **W2 & W3: Three-stage adaptation and latent-space integrity.**
> We appreciate this concern. In fact, the purpose of the three-stage adaptation procedure is to preserve the integrity of the pretrained latent space as much as possible while adapting to a new embodiment. Because the new embodiment comes from a substantially different distribution and the pretrained world model has never seen its dynamics, we first apply a very low-rank LoRA update (rank 2) to the pretrained world model in Stage 1. This allows only a slight shift toward the new embodiment while largely preserving the pretrained latent representations. In Stage 2, we freeze the world model and train only the action projector, so that the new embodiment’s raw actions are mapped into the pretrained latent action space rather than forcing the latent space itself to change. In Stage 3, we jointly finetune the action projector and FDM, again with a very low-rank LoRA update (rank 2), to better align the projected actions with the adapted dynamics model while still minimally perturbing the pretrained latent structure.
>
> Therefore, the three-stage procedure is designed not to compromise the pretrained latent space, but rather to retain it as much as possible during adaptation. This is also consistent with the strong generalization results after adaptation shown in Table 5. While the adaptation is performed in three stages, the total number of training iterations remains the same, so the overall efficiency of the framework is not reduced.
>
> **W4: Disjoint latent space and shared action encoder.**
> We thank the reviewer for raising this concern. To test whether the disjoint latent space is merely a consequence of using separate action encoders, **we additionally trained an EAC-WM variant with a shared action encoder across embodiments. We observed that the resulting action embeddings were still substantially disjoint across datasets.** While we cannot include the figure in the OpenReview comment, we will include the new figure in the revised version. This suggests that the fragmented latent space is not solely due to encoder separation, but is a more fundamental limitation of explicit-action conditioning under heterogeneous embodiments.
>
> **W5: Ablation of motion decoding and cross-augmentation.**
> We agree that Table 3 does not disentangle the individual contributions of motion decoding and cross-augmentation. Our intent in this ablation was to test whether removing both mechanisms that encourage physically meaningful latent actions harms downstream planning, and Table 3 shows that it does.

---

> > ### Author Rebuttal · Reviewer_8Mxt · 2026-04-03
> >
> > I appreciate the authors' clarifications. I will maintain my current score.

---

> > > ### Author Response · Authors · 2026-04-07
> > >
> > > We thank the reviewer for the positive feedback! We have added additional experiment on 5 Libero tasks to evaluate the LAM performance on a different task setting. Because the original Libero benchmark is close to saturated, we construct 5 more challenging variants by introducing unseen objects (book, white bowl, wine bottle) into the basket, or by changing the locations of seen objects (alphabet soup, salad dressing). For action selection, we use a pi0.5 policy trained with 50-step prediction as the VLA. At each decision step, we sample 20 candidate action sequences from the VLA and execute the first 10 steps. For each task, we evaluate 25 rollouts. The results are shown below.
> > >
> > > | Task | Random-Selected | Average-Selected | EAM | LAM |
> > >   |------|-----------------|-----------------|-----|-----|
> > >   | book | 76% | 76% | 80% | **92%** |
> > >   | white_bowl | 76% | 92% | 92% | **96%** |
> > >   | wine_bottle | 72% | 84% | 84% | **96%** |
> > >   | alphabet_soup | 80% | 72% | 80% | **88%** |
> > >   | salad_dressing | 68% | 84% | 76% | **88%** |
> > >   | **Average** | 74.4%| 81.6% | 82.4% | **92.0%** |
> > >
> > > As shown above, LAM consistently outperforms both EAM and the VLA baselines, including VLA with averaged actions. EAM provides only a modest improvement over the VLA baselines, whereas LAM yields a substantial gain, achieving an average success rate of 92.0%. Since these Libero tasks are easier than the RoboCasa setting, the strong performance here suggests that the relatively lower absolute success rate of LAM in RoboCasa is primarily constrained by the underlying VLA performance rather than by the quality of the learned world model itself.
> > >
> > > We hope this can further address your concern on incremental gains and we would really appreciate it if you can consider increasing your score.

---

### Official Review · Reviewer_kvvw · 2026-03-15

**Soundness:** 3
**Presentation:** 3
**Significance:** 3
**Originality:** 3
**Overall Recommendation:** 4
**Confidence:** 3

**Summary:**

This paper proposes a latent-action-conditioned robot world model for cross-embodiment learning. The key idea is to learn a unified latent action space shared across human and robot data, then adapt the pretrained model to unseen embodiments through an action projector. Experiments on imagined rollout and downstream planning suggest that the latent-action formulation transfers better than explicit action conditioning and scales more favorably with more pretraining embodiments.

**Compliance With Llm Reviewing Policy:**

Affirmed.

**Final Justification:**

The authors have addressed all my concerns. Thus, I decided to maintain the positive rating.

**Key Questions For Authors:**

How sensitive are the gains to the choice of pretraining datasets and embodiment mix?

Can the authors clarify which component contributes most: latent action learning, motion decoding, or the adaptation pipeline?

Do the same conclusions hold in real-world or partially real-world settings?

Can the authors discuss why downstream success remains low despite clear relative gains?

**Limitations:**

See Weakness

**Strengths And Weaknesses:**

The paper studies an important problem in robot world models: how to exploit heterogeneous cross-embodiment data despite incompatible action spaces. The method is sensible and the empirical results are generally supportive, with consistent gains over the explicit-action baseline in rollout quality and downstream planning. I also found the latent-space analysis and the scaling experiment across embodiments useful, as they support the main claim that a unified latent action space is beneficial.

The main weakness is that the novelty is moderate rather than major: the paper mainly combines latent action modeling, world models, motion decoding, and action projection into a cross-embodiment setting. In addition, the experimental validation is still somewhat limited: the downstream success rates remain fairly low, evaluation is mainly in simulation, and real-world validation is left to future work. The paper is technically solid enough for acceptance, but the overall impact is somewhat constrained by the limited task breadth and the lack of stronger deployment evidence.

---

> ### Author Rebuttal · Authors · 2026-03-31
>
> We thank the reviewer for the time and effort, and we appreciate the positive assessment that the paper is technically solid enough for acceptance.
>
> **W1: Moderate novelty and limited experimental validation.**
> We agree that many individual components of LAC-WM have appeared in prior work. Our contribution is not a new standalone module, but rather a practical and scalable framework for learning robot world models from cross-embodiment data. We believe the main contribution lies in the overall framework design and the empirical study. Across latent-space analysis, video generation quality, and downstream planning, LAC-WM consistently outperforms EAC-WM, a widely used explicit-action formulation. Since LAC-WM is a robot world model framework, we believe evaluation along these three axes provides a broad and meaningful validation of its benefits. We acknowledge that the downstream planning experiments are still limited in scope, and we will clarify this limitation more explicitly in the revised paper.
>
> **Q1: Sensitivity to pretraining dataset and embodiment mix.**
> We study this question in Figures 4 and 5. These experiments vary both the amount of pretraining data and the number of embodiments. The results show that once more than one embodiment is included during pretraining, LAC-WM consistently outperforms EAC-WM. More importantly, the performance gap widens as the number of embodiments and dataset size increase, indicating that LAC-WM scales more effectively in cross-embodiment settings.
>
> **Q2: Contribution of each component.**
> The three components play distinct but complementary roles. First, latent action learning enables cross-embodiment modeling by removing the dependence on incompatible explicit action spaces. Second, the motion decoder is critical for encouraging the latent actions to encode physically meaningful motion rather than relying on shortcut visual cues. Third, the adaptation pipeline provides a practical way to use a latent-action-conditioned world model at inference time for a new embodiment.
>
> **Q3: Real-world or partially real-world settings.**
> We agree that real-world validation is an important next step. In this work, we focus on establishing the benefits of the framework in realistic simulation environments. The RoboCasa environments are diverse and visually realistic, as also illustrated in the supplementary videos. In addition, Table 1 shows that LAC-WM produces substantially stronger video generation quality than EAC-WM.
>
> **Q4: Why downstream success remains low despite clear relative gains.**
> The absolute success rates remain low primarily because the evaluation setting is challenging. The tasks involve dexterous manipulation, which is already difficult, and our evaluation is conducted on unseen object instances and unseen categories, which further increases the generalization difficulty. In addition, the base VLA policy is trained on only 7k trajectories across 22 tasks, so its performance is limited to begin with. Since the world model is used to improve action selection on top of this base policy, the overall performance is still bounded by the strength of the underlying VLA. We will add this discussion to the revised paper.

---

> > ### Author Rebuttal · Reviewer_kvvw · 2026-04-02
> >
> > I appreciate the follow-up. My main concerns have been fully addressed.

---

> > > ### Author Response · Authors · 2026-04-03
> > >
> > > We thank the reviewer for the positive feedback! We have added additional experiment on 5 Libero tasks to evaluate the LAM performance on a different task setting. Because the original Libero benchmark is close to saturated, we construct 5 more challenging variants by introducing unseen objects (book, white bowl, wine bottle) into the basket, or by changing the locations of seen objects (alphabet soup, salad dressing). For action selection, we use a pi0.5 policy trained with 50-step prediction as the VLA. At each decision step, we sample 20 candidate action sequences from the VLA and execute the first 10 steps. For each task, we evaluate 25 rollouts. The results are shown below.
> > >
> > > | Task | Random-Selected | Average-Selected | EAM | LAM |
> > >   |------|-----------------|-----------------|-----|-----|
> > >   | book | 76% | 76% | 80% | **92%** |
> > >   | white_bowl | 76% | 92% | 92% | **96%** |
> > >   | wine_bottle | 72% | 84% | 84% | **96%** |
> > >   | alphabet_soup | 80% | 72% | 80% | **88%** |
> > >   | salad_dressing | 68% | 84% | 76% | **88%** |
> > >   | **Average** | 74.4%| 81.6% | 82.4% | **92.0%** |
> > >
> > > As shown above, LAM consistently outperforms both EAM and the VLA baselines, including VLA with averaged actions. EAM provides only a modest improvement over the VLA baselines, whereas LAM yields a substantial gain, achieving an average success rate of 92.0%. Since these Libero tasks are easier than the RoboCasa setting, the strong performance here suggests that the relatively lower absolute success rate of LAM in RoboCasa is primarily constrained by the underlying VLA performance rather than by the quality of the learned world model itself.
> > >
> > > We hope this can further address your concern on limited experimental validation and we would really appreciate it if you can consider increasing your score.

---

### Decision · Program_Chairs · 2026-04-30

**Decision:**

Accept (regular)

**Comment:**

This paper received ratings of 4, 4, 3, and 4, with a clear accept-leaning consensus. After rebuttal, three reviewers stated their concerns were adequately addressed and maintained positive recommendations; one reviewer remained weak reject, mainly due to limited task breadth and lack of real-world validation.

Reviewers agreed on the key strengths: important cross-embodiment problem, a clear latent-action framework, controlled comparisons against explicit-action conditioning, and consistent gains in rollout quality and downstream planning, including positive scaling with embodiment diversity. The main remaining limitation is scope (simulation-heavy evaluation and limited task diversity), rather than a fundamental methodological flaw.

After reviewing the paper, rebuttal, and discussion, the AC agrees with the majority consensus and sees no sufficiently compelling reason to overturn it.

Final Recommendation: Accept